# Serum Epiplakin Might Be a Potential Serodiagnostic Biomarker for Bladder Cancer

**DOI:** 10.3390/cancers13205150

**Published:** 2021-10-14

**Authors:** Soichiro Shimura, Kazumasa Matsumoto, Yuriko Shimizu, Kohei Mochizuki, Yutaka Shiono, Shuhei Hirano, Dai Koguchi, Masaomi Ikeda, Yuichi Sato, Masatsugu Iwamura

**Affiliations:** 1Department of Urology, Kitasato University School of Medicine, Sagamihara Campus, 1-15-1 Kitasato, Sagamihara 252-0374, Kanagawa, Japan; sou-telb89@live.jp (S.S.); yulico@med.kitasato-u.ac.jp (Y.S.); shiono-y@med.kitasato-u.ac.jp (Y.S.); s.hirano@med.kitasato-u.ac.jp (S.H.); dai.k@med.kitasato-u.ac.jp (D.K.); ikeda.masaomi@grape.plala.or.jp (M.I.); yuichi@med.kitasato-u.ac.jp (Y.S.); iwamura@med.kitasato-u.ac.jp (M.I.); 2International University of Health and Welfare Atami Hospital, 13-1 Higashikaigan Town, Atami 413-0012, Shizuoka, Japan; kmochi14338@iuhw.ac.jp

**Keywords:** bladder cancer, radical cystectomy, epiplakin, diagnosis

## Abstract

**Simple Summary:**

Improving early diagnosis and long-term postoperative monitoring of bladder cancer has become a focus of international research. In the present study, we evaluated the epiplakin expression levels in sera from patients with bladder cancer via a micro-dot blot array. Serum epiplakin levels were significantly higher in patients with bladder cancer than in those with stone disease and in healthy volunteers. Furthermore, serum epiplakin levels did not differ between patients with non-muscle-invasive and muscle-invasive bladder cancer. Immunohistochemistry revealed no association between staining scores, clinicopathological findings, and patients’ outcomes. In summary, our findings showed that serum epiplakin might be a potential diagnostic biomarker for patients with bladder cancer.

**Abstract:**

Tumor markers that can be detected at an early stage are needed. Here, we evaluated the epiplakin expression levels in sera from patients with bladder cancer (BC). Using a micro-dot blot array, we evaluated epiplakin expression levels in 60 patients with BC, 20 patients with stone disease, and 28 healthy volunteers. The area under the curve (AUC) and best cut-off point were calculated using receiver-operating characteristic (ROC) analysis. Serum epiplakin levels were significantly higher in patients with BC than in those with stone disease (*p* = 0.0013) and in healthy volunteers (*p* < 0.0001). The AUC-ROC level for BC was 0.78 (95% confidence interval (CI) = 0.69–0.87). Using a cut-off point of 873, epiplakin expression levels exhibited 68.3% sensitivity and 79.2% specificity for BC. However, the serum epiplakin levels did not significantly differ by sex, age, pathological stage and grade, or urine cytology. We performed immunohistochemical staining using the same antibody on another cohort of 127 patients who underwent radical cystectomy. Univariate and multivariate analysis results showed no significant differences between epiplakin expression, clinicopathological findings, and patient prognoses. Our results showed that serum epiplakin might be a potential serodiagnostic biomarker in patients with BC.

## 1. Introduction

Bladder cancer (BC) is one of the most common genitourinary tumors [1]. At initial diagnosis, most patients have non-muscle-invasive bladder cancer (NMIBC) and are treated with transurethral resection of the bladder tumor (TURBT) [2]. Up to 70% of NMIBC patients eventually relapse, and 10–20% experience disease progression to muscle-invasive bladder cancer (MIBC) [3]. Cystoscopy and urine cytology are typical modalities for diagnosing and surveilling BC. Cystoscopy helps detect tumor lesions but is painful and invasive even if flexible. Although urine cytology is less invasive, one of its limitations is low sensitivity [4]. Improving early diagnosis and long-term postoperative monitoring of BC have become a focus of international research. Some tumor markers (e.g., BTA and NMP22) may be useful for BC [5,6]. NMP22 is reported to have 32–92% sensitivity and 51–94% specificity; BTA is reported to have 51–94% sensitivity and 53–89% specificity. However, the clinical effectiveness of these markers is modest.

Specific genes have been investigated and evaluated in BC tissues, and we previously reported some BC-related proteins [7]. A comprehensive study of high molecular mass (HMM) protein expression in bladder cancer tissue was investigated by agarose two-dimensional gel electrophoresis followed by the analysis of liquid chromatography tandem mass spectroscopy. As a result of a literature search on the association between proteins’ expression and bladder cancer outcomes, eight proteins, including epiplakin, were selected for new biomarkers. These proteins were not previously reported in terms of any relation of bladder cancer. One candidate protein was epiplakin. Epiplakin is a 552-kDa protein originally identified as an autoantigen in the serum of a patient with a subepidermal blistering disease [8]. Epiplakin is involved in wound healing and mechanical skin strengthening [9,10] and is reported in the UALCAN database to be expressed in normal bladder tissue [11]. However, there are few articles that have shown a link between epiplakin and cancer, including BC.

Here, we assessed whether the dynamics of serum epiplakin could be used to diagnose BC and predict the outcomes in patients with BC. We conducted this study to investigate the circulating levels of epiplakin in sera from patients with BC, patients with stone disease, and healthy volunteers. We also assessed whether the serum epiplakin and immunohistochemical staining of surgical specimens would be associated with clinicopathological findings and patient prognoses.

## 2. Materials and Methods

### 2.1. Patients

We retrospectively analyzed 60 patients with BC who were treated at Kitasato University Hospital from March 2010 to September 2014. The study group comprised 46 men (77%) and 14 women (23%), with a median age of 74.5 years (range, 29–88 years). Fifty-five patients were treated via transurethral resection (TUR). Of the patients initially treated with TUR, 20 subsequently underwent radical cystectomy, and 4 underwent partial cystectomy. The five patients who did not undergo TUR were initially treated with radical cystectomy and bilateral pelvic lymphadenectomy.

Serum epiplakin levels were measured preoperatively. No anticoagulant was used in the measurement of serum epiplakin. Surgery was the initial treatment in patients with BC. Blood tests, chest X-rays, and computed tomography or magnetic resonance imaging were routinely performed, and no patients had distant metastases. Tumor-node-metastasis (TNM) staging was based on the 2002 TNM Classification of the International Union for Cancer Control and American Joint Committee on Cancer Guidelines [12]. Tumor grading was assessed according to the 1998 World Health Organization/International Society of Urologic Pathology consensus [13]. The median follow-up time was 51.3 months (mean: 46.0 months; range: 3.3–92.8 months) for those patients still alive at the last follow-up. No patients had previous radiation or systemic chemotherapy before surgical treatment, and none had histories of other cancers, skin diseases or pulmonary diseases.

We also measured serum epiplakin levels in 20 patients with stone disease and 28 healthy volunteers. The ethics committee of Kitasato University School of Medicine and Hospital (B17-010, B18-149) approved the study. All participants were treated as per the approved ethical guidelines. Patients could refuse entry or discontinue participation at any time.

### 2.2. Measurement of Serum Epiplakin

Patient and control sera were kept at −80 °C until use. Monoclonal antibodies specific for epiplakin were gifted from Drs. Tsuchisaka and Hashimoto [14]. Serum epiplakin was measured using reverse-phase protein array (RPPA) analysis [15,16]. Serum epiplakin levels were detected using an automated micro-dot blot array spotBot3 (Arrayit Corp., Sunnyvale, CA, USA). Serum samples were diluted 1:800 with 0.01% Triton X-100/phosphate-buffered saline (PBS) without bivalent ions and spotted onto high-density amino-group-induced glass slides with dimethyl sulfoxide (SDM0011; Matsunami Glass Ind., Ltd., Osaka, Japan). The glass slides were then blocked with 0.5% casein sodium (Wako Pure Chemical Industries, Osaka, Japan) for 1 h at room temperature (RT), then reacted with 400 times diluted rabbit anti-epiplakin antibody with 0.5% casein sodium for 2 h at RT. After being washed with 0.01% Triton X-100/PBS, the slides were incubated with biotinylated anti-rabbit IgG diluted 1:100 (BA-1000, Vector Laboratories, Burlingame, CA, USA) for 1 h at RT and diluted 1:1000 with streptavidin-horseradish peroxidase conjugate (GE Healthcare Bio-Sciences, Pittsburgh, PA, USA) for 30 min at RT. Peroxidase activity was detected using the Tyramide Signal Amplification Cyanine 5 System (PerkinElmer Life Sciences, Boston, MA, USA) diluted 1:100 for 20 min at RT. The slides were counterstained with Alexa Fluor 546-labeled goat anti-human IgG diluted 1:2000 (Life Technologies, Carlsbad, CA, USA) for 5 min at RT. Finally, the stained slides were scanned on a microarray scanner (Genepix 4000B; Molecular Devices, Sunnyvale, CA, USA). The fluorescence intensity, defined as the median net value of quadruple samples, was determined using the Genepix pro 6.0 software package (Molecular Devices).

Data were analyzed using DotBlotChip System software, version 4.0 (Dynacom Co., Ltd., Chiba, Japan). Normalized signals are presented as the positive intensity minus background intensity around the spot.

### 2.3. Immunohistochemistry and Scoring

We performed immunohistochemistry for radical cystectomized tissues using the same antibody in 127 consecutive cases at Kitasato University Hospital from October 1995 to June 2015. Paraffin-embedded 3 µm-thick sections of the harvested samples were deparaffinized in xylene, rehydrated in a descending ethanol series, and treated with 3% hydrogen peroxide for 15 min. After blocking with Protein Block Serum-Free (Agilent Technologies, Santa Clara, CA, USA) for 10 min, the sections were reacted with anti-epiplakin monoclonal antibody diluted 1:200 at RT for 1 h. After rinsing three times in Tris-buffered saline for 5 min each, the sections were incubated with Histfine Simple Stain MAX Peroxidase (Nichirei, Tokyo, Japan) at RT for 30 min. The sections were subsequently stained with stable DAB solution (Agilent Technologies) and counterstained with Mayer’s hematoxylin.

Immunohistochemistry was evaluated semiquantitatively by incorporating both the staining intensity and percentage of positive tumor cells (labeling frequency). The percentages of positive cells were scored as 0 for 0%, 1+ for 1–25%, 2+ for 26–50%, 3+ for 51–75%, or 4+ for 76–100%. The staining intensity was also scored as 1+ (weakly positive), 2+ (moderately positive), or 3+ (strongly positive). The multiply index was obtained by totaling the intensity and percentage scores. Epiplakin expression scores were stratified further as low (≤6) or high (>6) for the prognostic analyses. Two investigators (S.S. and Yuichi Sato) who were blinded to the clinical and pathological data reviewed all slides. Discordant cases were reviewed and discussed until a consensus was reached.

### 2.4. Statistical Analyses

The serum epiplakin levels between patients with BC and controls, including those with stone disease and healthy volunteers, and clinicopathological findings were compared via the analysis of variance and Mann–Whitney U test. The area under the curve (AUC) and best cut-off point were calculated using receiver-operating characteristic (ROC) analysis.

Association of the clinicopathological findings with immunohistochemistry of epiplakin expression was assessed using the chi-square test (or Fisher’s exact test, if appropriate) for categorical variables. Recurrence-free survival and cancer-specific survival were estimated with the log-rank test.

Univariate and multivariate analyses were performed using the Cox proportional hazards regression model, controlling for the effects of epiplakin and clinicopathological parameters. The statistical significance level was set at *p* < 0.05. Stata v. 16 for Windows (Stata, Chicago, IL, USA) was used for all analyses.

## 3. Results

### 3.1. Validation of Serum Epiplakin Levels

Figure 1 shows the serum epiplakin levels in patients with BC, those with stone disease, and healthy volunteers. Serum epiplakin levels were significantly increased in patients with BC compared with those with stone disease (*p* = 0.0013) and healthy volunteers (*p* < 0.0001). No significant differences were found between NMIBC and MIBC (*p* = 0.63). Patients with NMIBC also had significantly higher serum epiplakin levels than did those with stone disease (*p* = 0.0016) and healthy volunteers (*p* < 0.0001). No significant difference was found between patients with stone disease and healthy volunteers (*p* = 0.28).

ROC analysis was used to compare the serum epiplakin levels in BC patients with those of stone disease and healthy volunteers. The AUC for all BC patients was 0.78 (Figure 2a). Using an optimal cut-off point of 873 (95% confidence interval (CI) = 0.69–0.87) revealed that serum epiplakin levels exhibited 68.3% sensitivity and 79.2% specificity for BC. If specificity was increased, serum epiplakin levels for BC showed as follows: 40% sensitivity and 90% specificity, 31.7% sensitivity and 95% specificity, and 10% sensitivity and 98% specificity, respectively. The AUC for NMIBC patients only was 0.70 (Figure 2b). Using an optimal cut-off point of 1051 (95% CI = 0.69–0.89), serum epiplakin levels exhibited 64.7% sensitivity and 81.3% specificity for NMIBC.

### 3.2. Association of Serum Epiplakin Levels with Clinicopathological Characteristics

Table 1 shows the relationships between the serum epiplakin levels and clinicopathological features. Serum epiplakin levels did not significantly differ by sex, age, pathological stage and grade, or urine cytology.

### 3.3. Association of Serum Epiplakin Level with BC Recurrence

BC recurred in 13 of 34 patients (38%) with pT1 (*n* = 17) or less (*n* = 17) (median time to recurrence, 19.5 months; range, 5–44 months). Univariate and multivariate analyses for predicting BC recurrence demonstrated that epiplakin was not a significant factor (Appendix A).

### 3.4. Immunohistochemistry of Epiplakin

Epiplakin was expressed at various levels mainly in the tumor cell membrane and cytoplasm (Figure 3). Univariate and multivariate analyses revealed that epiplakin expression, clinicopathological findings, and patient outcomes did not significantly differ (Appendix A).

## 4. Discussion

Several studies have been conducted to find molecular markers that might identify progression from normal urothelium to BC. Both protein and gene analyses have frequently been conducted on BC tissues, and reports have suggested a relationship with mutagenesis and protein gene mutation [17]. However, no circulating molecular biomarkers or genetic mutations have been approved for clinical use. Here, we used a micro-dot blot array to show that serum epiplakin levels were higher in patients with BC than in those with stone disease and in healthy volunteers. However, no association was found between staining scores for BC tissue, clinicopathological findings, and patient prognoses. Furthermore, serum epiplakin levels did not differ between patients with NMIBC and those with MIBC. Hence, serum epiplakin might be a potential diagnostic marker in patients with BC, perhaps even in the early stages.

Plakins are a family of gigantic proteins that make up the cell cytoskeleton in cell-to-cell junctions mediated by cadherin. Some plakins are specific for cell junctions in the epithelium that are linked to intermediate filaments [8]. Plakins play various roles in cancer [18,19,20] and are correlated with urothelial cancer. We previously showed that periplakin expression was significantly correlated with the aggressive pathology and cancer-specific survival in patients with BC [21]. Kudo et al. reported that, in maxillary sinus cancer, p53-mutant tumors exhibited increased expression of cell adhesion genes containing epiplakin [22]. These data suggest that cell adhesion proteins are involved in cancer development and progression.

However, few articles have reported an association between epiplakin and cancer. Yoshida et al. reported immunohistochemical findings for epiplakin in pancreatic ductal adenocarcinoma precursor lesions [23] and demonstrated that epiplakin was expressed in precancerous lesions but not in pancreatic cancer. Dong et al. reported KLF5-mediated upregulation of epiplakin in tissues can activate the p38 signaling pathway to promote the proliferation of cervical cells [24]. Furthermore, this result suggested that epiplakin may be a target factor for the treatment of cervical cancer. Another possible mechanism of action is that epiplakin connects intermediate filaments (IFs) [25]. IF regulates a variety of cellular processes from cell migration to apoptosis and proliferation [26,27]. Keratin belongs to a family of IF proteins expressed in all epithelial cells and provides important structural support for mechanical and non-mechanical stress. Keratin is immunologically altered because cancer cells are derived from epithelial cells. Studies in epiplakin-deficient mice have shown that epiplakin plays a role in keratin filament rearrangement in response to stress [28,29]. Taken together, it is suggested that the association between epiplakin and carcinogenesis may be mediated by keratin. Our immunohistochemical study revealed no correlation between epiplakin expression and clinicopathological findings. However, to our knowledge, no study has reported a relationship between serum epiplakin levels and cancers. While serum epiplakin expression levels were not associated with clinicopathological findings or outcomes, these levels were significantly increased in patients with BC compared with those in patients with stone disease and in healthy volunteers. Although the mechanism for elevated serum epiplakin in BC and the relationship between immunohistochemical expressions in BC tissues are unclear, this study is the first to show that serum epiplakin might be a potential biomarker for diagnosing BC.

Cystoscopy and urine cytology are effective tools for diagnosing BC. Although flexible cystoscopy has made examinations easier for patients, it remains an invasive procedure. This study was conducted to investigate the circulating levels of epiplakin in sera as a potential diagnostic marker for BC. We found that epiplakin presented adequate sensitivity (64.7%) and specificity (81.3%) for NMIBC. Tilki et al. reported that the sensitivity and specificity of BTA ranged from 57–83% and 60–92%, respectively, and the sensitivity and specificity of NMP22 ranged from 47–100% and 60–90%, respectively. However, BTA and NMP22 are reported to yield high false-positive rates for urinary stones [30]. Although epiplakin showed similar sensitivity and specificity to those of BTA and NMP22, these markers differed significantly between patients with BC and those with stone disease. Thus, we think that serum epiplakin estimates might aid and overcome the problems of established BC markers.

Recent studies have reported that liquid biopsy by analysis of cell-free DNA (cfDNA) using next-generation sequencing (NGS) is useful [31]. Genetic panel assays for BCs such as Uroseek are also in development [32,33]. Patrice et al. reported that the sensitivity and specificity for detection for BCs using the singleplex assay UroMuTERT, which detects mutations in the promoter of the telomerase reverse transcriptase gene (TERT), was very high [34]. However, although NGS is more reproducible and accurate than the enzyme-linked immuno solvent assay (ELISA), it is expensive and requires a high degree of expertise and a high level of experimental equipment [35]. We think that micro-dot arrays are one of the simple methods. Compared to other cancers, the field of biomarker in bladder cancer is still insufficient. Although epiplakin currently has many issues to be used in daily clinical practice, it could be a promising biomarker.

This study has several limitations. First, epiplakin is located not only in the urothelium but in the epithelium and other organs [11]. Second, the role of serum epiplakin expression must be validated in other diseases, including cancerous and inflammatory lesions. One study reported that plakin deletion in the lungs caused overexpression of anti-inflammatory cytokines [36]. In terms of urinary tract infection, epiplakin was not measured in this study. It invalidates biomarker results because it can cause false-positive findings. When screening with biomarkers, people with urinary tract infection have to be treated first for their infection before they can be re-examined with cancer biomarkers. However, differentiation from urinary tract infection is still one of the issues. Third, although we performed immunohistochemical staining of BC tissues, the patient cohorts differed between the serum and immunohistochemistry groups. The dynamics of epiplakin between serum and immunohistochemistry findings must be determined. Fourth, the serum levels of epiplakin were only measured using micro-dot blot analysis. It will be necessary to compare with other measurement methods such as Western blotting and ELISA, considering the stable measurement of epiplakin in clinical application. Finally, serum epiplakin was not associated with BC recurrence on multivariate analysis. Serum epiplakin levels at the time of diagnosis may reflect only existence of a tumor but not the biological aggressiveness of the tumor. At the moment, epiplakin is a possible candidate that could be tested in a study for their ability to detect bladder cancer.

## 5. Conclusions

In conclusion, patients with BC demonstrated significantly increased serum epiplakin expression compared with those in patients with stone disease and in healthy volunteers. Multi-institutional evaluations of serum epiplakin in a large patient population are warranted before serum epiplakin can be included as a biomarker for routine clinical use for early diagnosis of BC. Serum epiplakin levels might be a suitable non-invasive diagnostic method as an adjunct to urinary cytology and cystoscopy for diagnosis of bladder cancer once proven in a prospective cohort study.

## 6. Patents

Epiplakin has been patented in Japan as a diagnostic marker for bladder cancer (JP. PAT. 6060425). The Kitasato Institute, Diagnosis of Bladder Cancer, Japan, No. 6060425.

## Figures and Tables

**Figure 1 cancers-13-05150-f001:**
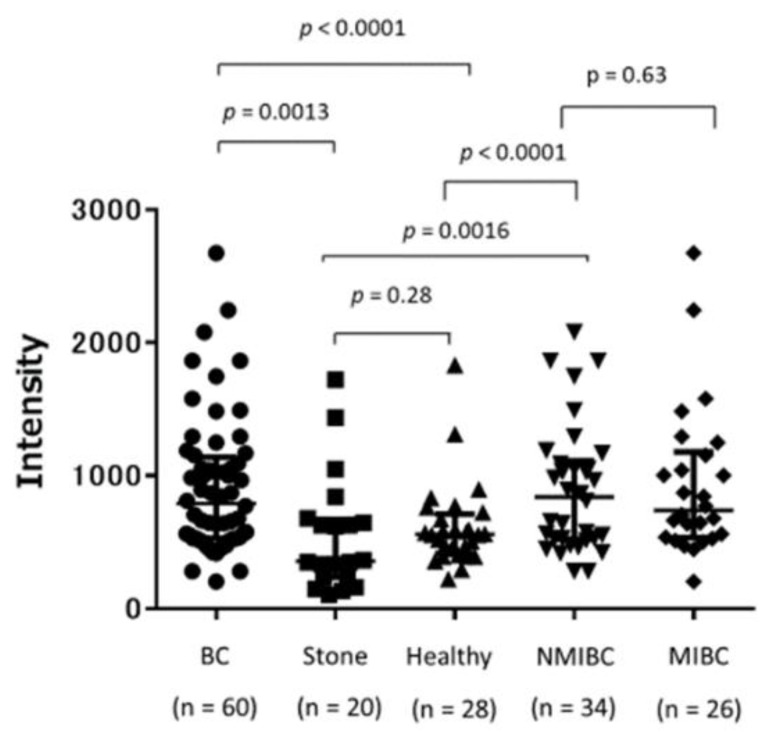
Serum epiplakin expression levels in patients with stone disease, healthy volunteers, and patients with bladder cancer. Serum epiplakin levels were significantly higher in patients with bladder cancer than in those with stone diseases and in healthy volunteers. Serum epiplakin levels were also significantly higher in patients with NMIBC than in those with stone diseases and in healthy volunteers. NMIBC: non-muscle-invasive bladder cancer, MIBC: muscle-invasive bladder cancer.

**Figure 2 cancers-13-05150-f002:**
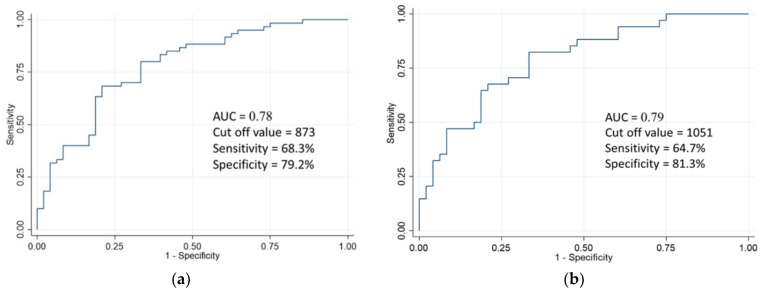
Receiver-operating characteristic (ROC) analysis of serum epiplakin expression levels (**a**) ROC analysis of serum epiplakin expression levels in patients with bladder cancer. The area under the ROC curve level was 0.78. The sensitivity and specificity were 68.3% and 79.2%, respectively, using a cut-off point of 873. (**b**) ROC analysis of serum epiplakin expression levels in patients with NMIBC. The area under the curve-ROC level for bladder cancer was 0.79. The sensitivity and specificity were 64.7% and 81.3%, respectively, using a cut-off point of 1051.

**Figure 3 cancers-13-05150-f003:**
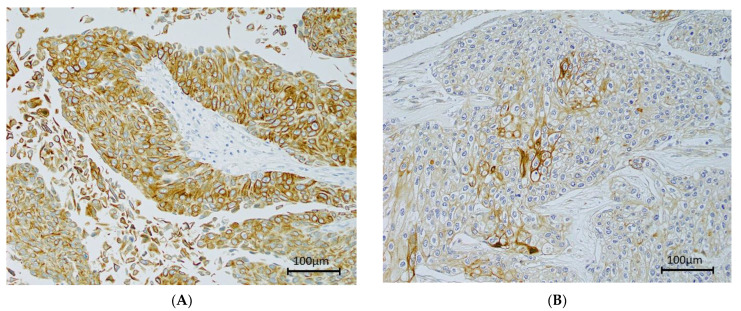
Immunohistochemical staining of epiplakin expression in bladder cancer. Microscopic images are representative tumor cells (200× magnification). (**A**) High expression (score = 12); (**B**) Low expression (score = 1).

**Table 1 cancers-13-05150-t001:** The relationships between of serum epiplakin levels and clinicopathological features.

Characteristics	No. of Pts.	Serum Epiplakin Level			*p*-Value
		Median	Mean	Range	
Age(years)					0.11
≤65	11	761.8	1136.6	373.5–3209	
>65	49	1302.8	1844.2	263.8–4955	
Sex					0.28
Male	45	1192.8	1813.4	346–6485.8	
Female	14	1181	1384	263.8–3501	
T stage					0.50
≤pT1	34	1250.9	1806.1	346–6845.8	
≥pT2	26	1175.6	1575.2	263.8–4955	
Grade					0.40
G1/2	32	1430	1849.5	383.5–6845.8	
G3	28	1177	1542.1	263.8–4955	
Urine cytology					0.51
<classⅢb	16	1171	1544.7	373.5–3982	
≥classⅢb	43	1303	1800	263.8–6845.8	

No.: number. pts.: patients.

## Data Availability

The datasets used and/or analyzed during the current study are available from the corresponding author on reasonable request.

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
