# Peer review of "Serum Epiplakin Might Be a Potential Serodiagnostic Biomarker for Bladder Cancer"

_cancers, 2021, doi:10.3390/cancers13205150_

Round 1
Reviewer 1 Report
It is important to find biomakers to predict the condition of cancers. Your finding will be a good marker to predict the bladder cancer. The AUC of ROC was high level to distinguish the patients or not. Combination with some biomaker allows to increase the precision accuracy.
Author Response
It is important to find biomakers to predict the condition of cancers. Your finding will be a good marker to predict the bladder cancer. The AUC of ROC was high level to distinguish the patients or not. Combination with some biomaker allows to increase the precision accuracy.
Answer : Thank you so much for your valuable comment. As you pointed out, epiplakin alone may not be a sufficient marker for bladder cancer. In the next study, we think it is necessary to measure the detection rate of early-stage bladder cancer in combination with non-invasive tests such as cytology or ultrasound. We will use your opinion for our future research.

Reviewer 2 Report
- Line 42: “present” is an adj. or n. (grammar error)
- Line 51: modist? Modest?
- Line 53-54: More information about why these eight proteins were selected is needed. Similarly, what is the reason why epiplakin was selected. Were the other seven proteins tested too? If yes, what are the results?
- Line 66-148: Detailed information about the methods used was provided. It is very helpful for readers who are interested in the experiments done in the paper.
- Line 75: More information about whether anticoagulant was used during the blood collection procedure will be helpful for readers who want to repeat or do similar research.
- Line 91: Only one method of measurement of epiplakin levels was used. To confirm the levels of epiplakin, at least another method needs to be done, especially when considering that micro-dot blot is fast but that accurate compared to other methods such as western blot.
- Line 95: What is the physiological concentration of epiplakin in the blood? Will the results still be accurate after a high dilution like 1:800?
- Line 139: To make sure the assumptions are met, the homogeneity of variance needs to be tested before conducting Mann-Whitney U test.
- Line 195: In Figure 3, a bar is needed in the immunohistochemical staining images.
- In the discussion part, the authors mentioned that there’s no evidence showing the relationship between serum epiplakin levels and cancers. If they are not even related, what makes epiplakin a convincing marker for bladder cancer? The authors may want to provide more citations about how epiplakin goes to the blood and what the potential roles of epiplakin is in bladder cancer.
- This paper provided evidence that serum epiplakin levels is a potential serodiagnostic biomarker of bladder cancer. Patients with stone disease were included to test whether epiplakin is a more accurate biomarker when compared to known markers such as NMP22 and BTA. The experimental design is logical. However, the levels of epiplakin were only tested by a single method -- micro-dot blot. Clinically, micro-dot blot is a convenient approach to test epiplakin levels, but in this paper, other experimental methods are needed to confirm the results of epiplakin levels from micro-dot blot.
Reviewer 3 Report
General:
Shimura et al. present a well written manuscript on the detection of the protein epiplakin in serum as a possible biomarker for bladder cancer. This is a new finding that has the potential to improve the performance of the currently used urine-based markers for bladder cancer. However, an important control group is missing in the presented experiments, which limits the scope of the findings. Also, the discussion of serum epiplakin in the context of other known biomarkers could be improved.
Major:
Urinary bladder infections are a major confounder of some urine-based biomarkers like NMP22 and BTA. Unfortunately, the controls did not include a group of patients with urocystitis (as already mentioned in the discussion). It could be a major advantage of serum-based markers like epiplakin in comparison to most of the urine-based markers, which might be more prone to local infections or other confounders like microhematuria. Is it possible to provide at least a small group of patients with bladder infections? Otherwise, epiplakin might be just another candidate in the long series of “promising” biomarkers.
At various places in the text and in the title the authors mention that epiplakin could be used as a potential marker for early detection of bladder cancer (lines 2, 20, 35, 207, 262). However, the study does not present any data to support this at all. For example, you don’t see a difference between ≤pT1 an ≥pT2. Also, the study design used is not suitable to address this problem. Please tone down the suggestion (especially in the title) that epiplakin might be a marker for early diagnosis.
Only a prospective study design with repeated serial sampling (or a sufficient number of chance findings of early stages) would be able to prove that a marker could be suitable to replace or supplement cytology or cystoscopy. I suggest reading the paper by Pesch et al. on study designs (Biochimica et Biophysica Acta 1844 (2014) 874-883). At the moment, epiplakin is just a possible candidate that could be tested in a study that is designed to test markers for their ability to detect bladder cancer early.
It is true that the clinical use of biomarkers for bladder cancer is currently rather modest. But your claim is based on just two (older) markers and publications that are more than 10 years old. There have been new developments. You should have a look at more recent reviews. You might use and cite one or two of the following papers: Batista et al., Diagnostics 2020, 10, 39; doi:10.3390/diagnostics10010039; Ferro et al., J. Pers. Med. 2021, 11, 237. https://doi.org/10.3390/jpm11030237; Avogbe et al., EBioMedicine 44 (2019) 431-438, https://doi.org/10.1016/j.ebiom.2019.05.004;
Please discuss in more detail the current state of biomarker research for bladder cancer and the possible roles of epiplakin in that context (e.g., combination with other markers as suggested and shown in other studies before).
Minor:
line 51: “modest” instead of “modist”
line 58: “…no association has been reported between epiplakin and malignant lesions…” This is indeed true for bladder cancer but as you mention in lines 218 ff, epiplakin plays a role in (pre)pancreatic cancer (citation [21]). You might have a look at a more recent paper by Ma et al. (KLF5-mediated Eppk1 expression promotes cell proliferation in cervical cancer via the p38 signaling pathway. BMC Cancer. 2021 Apr 8;21(1):377. doi: 10.1186/s12885-021-08040-y).
line 97: “amino-group-induced”? Do you mean functionalized with amino groups (like coated with aminosilane)?
line 114: How did you quantify binding of epiplakin. Did you use a standard curve with known concentrations? Is it possible with a dot blot assay to obtain quantitative results or is it rather semi-quantitative?
lines 158 ff: You only used the “best cut-off point” (Youden index?) resulting in a sensitivity of 68.3% and a specificity of 79.2%. When screening for cancer, it is important keep the number of false-positive results low in order to avoid unnecessary stress for the patients. Please provide also the sensitivities for some higher specificities that have been set to, e.g., 90%, 95%, and 98%.
line 190 ff.: Supplementary Table 4 is not mentioned in the main text, it could be included here.
line 210: Do you mean “…are specific for cell junctions…”?
line 262: “…may help physicians make treatment decisions…” I don’t see any indication for this in your study. Epiplakin does not show any correlation with clinical parameters like survival (Suppl. Figure S1 & S2”) or treatment (not tested).
You tested epiplakin in serum as well as bladder cancer tissue and demonstrated some differences between serum and tissue. Did you also test corresponding urine samples of the participants of the different study groups?
Author Response
Major
#1. Urinary bladder infections are a major confounder of some urine-based biomarkers like NMP22 and BTA. Unfortunately, the controls did not include a group of patients with urocystitis (as already mentioned in the discussion). It could be a major advantage of serum-based markers like epiplakin in comparison to most of the urine-based markers, which might be more prone to local infections or other confounders like microhematuria. Is it possible to provide at least a small group of patients with bladder infections? Otherwise, epiplakin might be just another candidate in the long series of “promising” biomarkers.
Answer: Thank you for your comment. It is true that bladder infections can be one of the major problems in the development of urine-based biomarkers. However, urinary infection itself may be simply detected in urine examination using urine sediment and in established serum inflammation markers. One of the clinical problems is microhematuria which you also pointed out, especially for patients with non-muscle invasive cancer which was relatively difficult to detect it by conventional urinary markers. According to clinical questions and limitations, we investigated serum level of epiplakin in patients with stone disease which was easily to show microhematuria. In the present study, epiplakin levels were higher in patents with bladder cancer than in those with stone disease. In addition, there is no significant difference between patients with non-muscle invasive cancer and those with muscle invasive cancer. We think it may be reasonable answer to detect bladder cancer using serum epiplekin. We agree with your comment, however, unfortunately we are unable to provide serum data of epiplakin in patients with bladder infections so that we simply described the limitation of this study. Thank you for your understanding and valuable comments.
Line 275
Second, the role of serum epiplakin expression must be validated in other diseases, including cancerous and inflammatory lesions. One study reported that plakin deletion in the lungs caused overexpression of anti-inflammatory cytokines. In terms of urinary infection, it may be simply detected in urine examination using urine sediment or in established serum inflammation markers.
#2. At various places in the text and in the title the authors mention that epiplakin could be used as a potential marker for early detection of bladder cancer (lines 2, 20, 35, 207, 262). However, the study does not present any data to support this at all. For example, you don’t see a difference between ≤pT1 and ≥pT2. Also, the study design used is not suitable to address this problem. Please tone down the suggestion (especially in the title) that epiplakin might be a marker for early diagnosis. Only a prospective study design with repeated serial sampling (or a sufficient number of chance findings of early stages) would be able to prove that a marker could be suitable to replace or supplement cytology or cystoscopy. I suggest reading the paper by Pesch et al. on study designs (Biochimica et Biophysica Acta 1844 (2014) 874-883). At the moment, epiplakin is just a possible candidate that could be tested in a study that is designed to test markers for their ability to detect bladder cancer early.
Answer: Thank you for your fruitful opinion. We think that it would be important that epiplakin did not differ between patient with ≤pT1 and those with ≥pT2. If the patient had bladder cancer, epiplakin itself was high in serum, irrespective of tumor stage. However, as you pointed it out, this study only showed that epiplakin might be a marker for early diagnosis. According to your suggestions, we changed the title and amended words from “may” to “might” in several sentences. And we described limitation more in discussion section. We agree with your comments that only a prospective study design with repeated serial sampling would be able to prove that a marker could be suitable to replace or supplement cytology or cystoscopy. Thank you for your valuable comments.
Title
Serum epiplakin might be a potential early serodiagnostic biomarker for bladder cancer
Line 286
Finally, serum epiplakin was not associated with BC recurrence on multivariate analysis. Serum epiplakin levels at the time of diagnosis may reflect only existence of a tumor but not the biological aggressiveness of the tumor. At the moment, epiplakin is just a possible candidate that could be tested in a study that is designed to test markers for their ability to detect bladder cancer early.
#3. It is true that the clinical use of biomarkers for bladder cancer is currently rather modest. But your claim is based on just two (older) markers and publications that are more than 10 years old. There have been new developments. You should have a look at more recent reviews. You might use and cite one or two of the following papers: Batista et al., Diagnostics 2020, 10, 39; doi:10.3390/diagnostics10010039; Ferro et al., J. Pers. Med. 2021, 11, 237. https://doi.org/10.3390/jpm11030237; Avogbe et al., EBioMedicine 44 (2019) 431-438, https://doi.org/10.1016/j.ebiom.2019.05.004; Please discuss in more detail the current state of biomarker research for bladder cancer and the possible roles of epiplakin in that context (e.g., combination with other markers as suggested and shown in other studies before).
Answer: Thank you so much for your advice. We read the literatures you gave us. We added and amended the discussion. We think and believe this method and its outcome will be one of the clues to test serum samples in patients with bladder cancer.
Line 262
Recent studies have reported that liquid biopsy by analysis of cell-free DNA (cfDNA) using next-generation sequencing (NGS) is useful [24]. Genetic panel assays for BCs such as Uroseek are also in development [25,26]. Patrice et al. reported that the sensitivity and specificity for detection for BCs using the singleplex assay UroMuTERT, which detects mutations in the promoter of the telomerase reverse transcriptase gene (TERT), was very high [27]. However, although NGS is more reproducible and accurate than the Enzyme-Linked Immuno Solvent Assay (ELISA) method, it is expensive and requires a high degree of expertise and a high level of experimental equipment [28]. We think that micro-dot arrays are one of the simple methods. Compared to other cancers, the field of biomarker in bladder cancer is still insufficient. Epiplakin currently has many issues to be used in daily clinical practice, therefore it is necessary to develop improved methods with high sensitivity and specificity in the future.
Minor:
#4. line 51: “modest” instead of “modist”
Answer: We have corrected it in the text.
Line 50
However, the clinical effectiveness of these markers is modest.
#5. line 58: “…no association has been reported between epiplakin and malignant lesions…” This is indeed true for bladder cancer but as you mention in lines 218 ff, epiplakin plays a role in (pre)pancreatic cancer (citation [21]). You might have a look at a more recent paper by Ma et al. (KLF5-mediated Eppk1 expression promotes cell proliferation in cervical cancer via the p38 signaling pathway. BMC Cancer. 2021 Apr 8;21(1):377. doi: 10.1186/s12885-021-08040-y).
Answer: Thank you so much for your useful comment. We have read several papers which you suggested, and changed and added the sentences.
Line 62
However, there are few articles that have shown a link between epiplakin and cancer, including BC.
Line 229
Dong et al. reported KLF5-mediated upregulation of epiplakin in tissues can activate the p38 signaling pathway to promote the proliferation of cervical cells [24]. Furthermore, this result suggested that epiplakin may be a target factor for the treatment of cervical cancer. Another possible mechanism of action is that epiplakin connects intermediate filaments (IFs) [25]. IF regulates a variety of cellular processes, from cell migration to apoptosis and proliferation [26, 27]. Keratin belongs to a family of IF proteins expressed in all epithelial cells and provides important structural support for mechanical and non-mechanical stress. Keratin is immunologically altered because cancer cells are derived from epithelial cells. Studies in epiplakin-deficient mice have shown that epiplakin plays a role in keratin filament rearrangement in response to stress [28, 29]. Taken these together, it is suggested that the association between epiplakin and carcinogenesis may be mediated by keratin.
#6. line 97: “amino-group-induced”? Do you mean functionalized with amino groups (like coated with aminosilane)?
Answer: The slide glass we used was a high-density amino group-introduced type glass originally developed by Matsunami Glass. "Amino-group-induced" means a coat that is more fixed and stable than general aminosilanes. However, I don't know the details because it may be protected by patent. For sure, the URL of the Matsunami Glass homepage is demonstrated.
https://www.matsunami-glass.co.jp/product/bio/glass_slide_for_dna_microarry/
#7. line 114: How did you quantify binding of epiplakin. Did you use a standard curve with known concentrations? Is it possible with a dot blot assay to obtain quantitative results or is it rather semi-quantitative?
Answer: Epiplakin was analyzed by the semi-quantitative method by reverse-phase protein array (RPPA) analysis which we established this method previously [1, 2]. The intensity score in Figure 1 has no unit. We do not measured a standard curve with known concentrations of epiplakin. However, we think that the light intensity can purely reflect the amount of serum epiplakin, because the sera of the 60 patients were reacted on the same plate. We added the sentence that RPPA analysis using micro-dot array was performed in the text.
Line 97
Serum epiplakin was measured using reverse-phase protein array (RPPA) analysis[15,16].
Serum epiplakin levels were detected using an automated micro-dot blot array spotBot3 (Arrayit Corp., Sunnyvale, CA, USA).
#8. lines 158: You only used the “best cut-off point” (Youden index?) resulting in a sensitivity of 68.3% and a specificity of 79.2%. When screening for cancer, it is important keep the number of false-positive results low in order to avoid unnecessary stress for the patients. Please provide also the sensitivities for some higher specificities that have been set to, e.g., 90%, 95%, and 98%.
Answer: Thank you for your sharp opinion. As you say, it is true that the sensitivity of epiplakin decreases considerably when specificity is increasing. We added the sentences in results section.
Line 166
Using an optimal cut-off point of 873 (95% confidence interval [CI] = 0.69–0.87) revealed that serum epiplakin levels exhibited 68.3% sensitivity and 79.2% specificity for BC. Serum epiplakin levels showed as followed: 40% sensitivity and 90% specificity, 31.7% sensitivity and 95% specificity, and 10% sensitivity and 98% specificity, respectively.
#9. line 190: Supplementary Table 4 is not mentioned in the main text, it could be included here.
Answer: As you described, we missed Supplementary Table 4 in the text. We added it below.
Line 202
(Supplementary Tables 2, 3 and 4; Supplementary Figures 1 and 2)
#10. line 210: Do you mean “…are specific for cell junctions…”?
Answer: Yes. We have changed the sentence.
Line 218
Some plakins are specific for cell junctions in the epithelium that are linked to intermediate filaments [8].
#11. line 262: “…may help physicians make treatment decisions…” I don’t see any indication for this in your study. Epiplakin does not show any correlation with clinical parameters like survival (Suppl. Figure S1 & S2”) or treatment (not tested).
Answer: Epiplakin did not correlate with prognosis or treatment in this study which you pointed out. We amended the sentence.
Line 297
Serum epiplakin levels might be a suitable non-invasive diagnostic method as an adjunct to urinary cytology and cystoscopy for early diagnosis of bladder cancer.
#12. You tested epiplakin in serum as well as bladder cancer tissue and demonstrated some differences between serum and tissue. Did you also test corresponding urine samples of the participants of the different study groups?
Answer: No. Currently, our research team is aiming to establish an experimental method for studying various substances from exosomes involving epiplakin in urine samples. Thank you for your valuable comments.
References
[1] Kobayashi, M.; Nagashio, R.; Jiang, S.X.; Saito, K.; Tsuchiya, B.; Ryuge, S.; Katono, K.; Nakashima, H.; Fukuda, E.; Goshima, N.; et al. Calnexin Is a Novel Sero-Diagnostic Marker for Lung Cancer. Lung Cancer, 2015, 90, 342-345.
[2] Yanagita, K.; Nagashio, R.; Jiang, S.X.; Kuchitsu, Y.; Hachimura, K.; Ichinoe, M.; Igawa, S.; Nakashima, H.; Fukuda, E.; Goshima, N.; et al. Cytoskeleton-Associated Protein 4 Is a Novel Serodiagnostic Marker for Lung Cancer. Am J Pathol, 2018, 188, 1328-1333.
Round 2
Reviewer 2 Report
The comments I pointed out were addressed in the revised manuscript, I don't have more comments.
Author Response
#1. The comments I pointed out were addressed in the revised manuscript, I don't have more comments.
Answer: Thank you for your comment.
Reviewer 3 Report
The authors have addressed all questions and comments. However, there are still some minor points:
- I am still not fully satisfied with the handling of the claim that epiplakin might be a marker for “early” detection. It would be better to just leave out the “early” (lines 2, 20, 35, & 311) and just state that epiplakin is “a potential serodiagnostic biomarker” – because there is no prove for early diagnosis.
Line 221: please insert “perhaps” before the “even”
Line 312-313: at the end of the sentence, please add (for example) “…once proven in a prospective cohort study”
- Lines 284-285: There might be a misunderstanding of my previous remarks: Cystitis is a confounder of many urine-based biomarkers, i.e., it invalidates biomarker results because it can cause false-positive results. Cystitis can be quite frequent in elderly people. When screening with biomarkers, people with bladder infections have to excluded and treated first for their infection before they can be re-examined with cancer biomarkers.
- Please check the grammar in the newly added text sections
Author Response
#1. I am still not fully satisfied with the handling of the claim that epiplakin might be a marker for “early” detection. It would be better to just leave out the “early” (lines 2, 20, 35, & 311) and just state that epiplakin is “a potential serodiagnostic biomarker” – because there is no prove for early diagnosis.
Line 221: please insert “perhaps” before the “even”
Line 312-313: at the end of the sentence, please add (for example) “…once proven in a prospective cohort study”
Answer: Thank you for your comment. I deleted the “early”, inserted “perhaps” before the “even”, and added “… once proven in a prospective cohort study” at the end of the conclusion.
Line 2
Serum epiplakin might be a potential serodiagnostic biomarker for bladder cancer
Line 20
In summary, our findings showed that serum epiplakin might be a potential diagnostic biomarker for patients with bladder cancer.
Line 35
Our results showed that serum epiplakin might be a potential serodiagnostic biomarker in patients with BC.
Line 216
Hence, serum epiplakin might be a potential diagnostic marker in patients with BC, perhaps even in the early stages.
Line 296
Serum epiplakin levels might be a suitable non-invasive diagnostic method as an adjunct to urinary cytology and cystoscopy for diagnosis of bladder cancer once proven in a prospective cohort study
#2. Lines 284-285: There might be a misunderstanding of my previous remarks: Cystitis is a confounder of many urine-based biomarkers, i.e., it invalidates biomarker results because it can cause false-positive results. Cystitis can be quite frequent in elderly people. When screening with biomarkers, people with bladder infections have to excluded and treated first for their infection before they can be re-examined with cancer biomarkers.
Answer: Thank you for useful comment. As you say, this study did not measure serum epiplakin in patients with cystitis, so there was a possibility of false positive. We corrected the sentences according to your comments.
Line 276
In terms of urinary tract infection, epiplakin was not measured in this study. It invalidates biomarker results because it can cause false-positive findings. When screening with biomarkers, people with urinary tract infection have to be treated first for their infection before they can be re-examined with cancer biomarkers. However, differentiation from urinary tract infection is still one of the issues.
#3. Please check the grammar in the newly added text sections.
Answer: Thank you for your comment. We checked the grammar in the newly added sentences.